# BrailleVision: Text Instruction Tuning of LLMs to Improve Visual Skills

## Abstract

Large Language Models (LLMs) have shown exceptional proficiency in natural language processing tasks. More recently, their potential is being explored in vision-centric applications. Current multimodal large language models (MLLMs) incorporate general-purpose LLMs through multimodal instruction tuning. These LLMs, however, lack prior vision centric text based training, potentially limiting their effectiveness. In this work, we propose a novel approach to enhance vision-related capabilities of general-purpose LLMs through instruction fine-tuning with vision-centric text data. Specifically, we curate a diverse dataset, BrailleVision-360K, to teach skills such as visual perception, abstraction, and spatio-temporal reasoning without the use of visual data, analogous to how Braille codes are used by the visually impaired. The dataset is constructed in an automated manner by utilizing LLMs, bootstrapping from existing datasets, and employing VLMs to improve quality. Next, to fine-tune an LLM with this dataset, we introduce Fine-SFT, a novel fine-tuning approach that improves upon standard supervised fine-tuning and preference optimization techniques. Our vision-specialized LLM shows significant performance gains in tasks such as visual classification and open vocabulary detection. Furthermore, when used as the '*backbone*' for an MLLM, our model outperforms existing LLMs on standard visual QA benchmarks while reducing hallucinations, highlighting the importance of vision-centric pretraining of LLMs in multimodal tasks.

## 1 Introduction

Large Language Models (LLMs) exhibit remarkable proficiency across diverse language understanding and generation tasks Minaee et al. (2024). This broad generalization has increasingly motivated their adoption in computer vision. Two high-level approaches have emerged for utilizing LLMs in vision tasks: first is extending LLMs to understand visual inputs and/or generate visual outputs. An example of this approach is multi-modal LLMs (MLLMs), like LLaVA Liu et al. (2024), BLIP-2 Li et al. (2023a), etc., which incorporates both text and visual input into its instruction-tuning dataset. In this setup, a general-purpose LLM is trained with multi-modal data to equip it with vision capabilities. The second approach combines an LLM with a Vision-Language Model (VLM) Radford et al. (2021); Jia et al. (2021); Zhai et al. (2023b). This approach relies on the LLM for its world knowledge and reasoning capabilities and VLM for its visual recognition capabilities. This approach has been utilized in tasks like visual classification Menon & Vondrick (2023), open vocabulary object detection Kaul et al. (2023), and Auto-Vocabulary segmentation Ülger et al. (2024). A key characteristic shared by both approaches is their use of LLMs trained on text data from generalized domains covering various topics. However, these LLMs lack specific prior adaptation for vision tasks, potentially limiting their effectiveness.

LLM training typically involves two key stages: large-scale pre-training (PT) and supervised fine-tuning with instruction following data (IFT). This dual-stage process allows LLMs to acquire vast general knowledge and unlock their capabilities for specific tasks through targeted instruction tuning Chung et al. (2024); Ouyang et al. (2022); Wang et al. (2023b). Instruction tuning, particularly with machine-generated instruction-following data, has significantly enhanced the zero-shot capabilities of LLMs on new tasks,showcasing a form of generalized intelligence. Another notable advantage of instruction tuning is its ability to allow large models to quickly adapt to specific domains or acquire specialized knowledge without requiring extensive computational resources or ma-

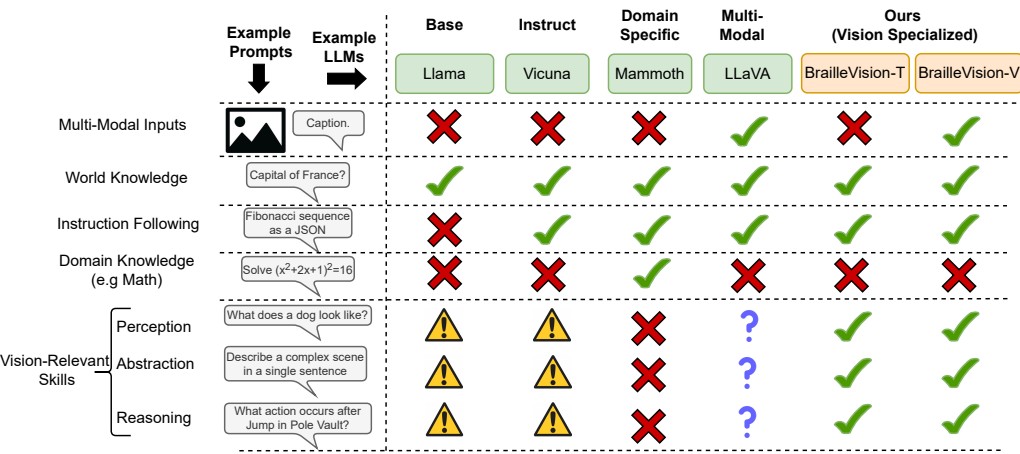

Figure 1: Although LLMs are general purpose models, different classes of LLMs specialized in different capabilites. Base LLMs are pre-trained on large-scale web corpora and possess a vast amount of world knowledge as a result. However, their ability to follow instructions must be unlocked through instruction fine-tuning. Domain specific LLMs are instruction tuned to answer prompts from a specific domain, e.g. math. Multimodal LLMs have an aligned input visual encoder to accept image/video inputs. This work focuses on vision-related skills such as visual perception, abstraction, and reasoning. While general-purpose LLMs exhibit some degree of visual reasoning, these abilities remain limited; in Multi-Modal LLMs these skills are partially learned during visual alignment and multi-modal fine-tuning. We propose BrailleVision, a text instruction tuning dataset designed to unlock these vision relevant skills in LLMs. Additionally, we also align a visual encoder with our LLM to produce BrailleVision-V, a MLLM with enhanced vision relevant skills.
Legend: ✖→ missing; ✓→ present; ⚠→ limited; ?→ partially learned

jor architectural changes. This adaptability is crucial for applications that require domain-specific expertise, as it facilitates rapid and efficient model customization. Furthermore, domain-specific instruction-tuned LLMs demonstrate improved alignment abilities to the target task, often outperforming proprietary LLMs in instruction adherence and output relevance. These domain specialized LLMs Ling et al. (2023) have achieved promising results in fields as diverse as Math Liu & Low (2023); Yue et al. (2023b); Roziere et al. (2023); Luo et al. (2023b), Medicine Li et al. (2023c), Legal Chalkidis et al. (2020) and Finance Wu et al. (2023). However, such an approach remains largely unexplored in vision-related applications. Current applications of LLMs in vision domain are limited to utilizing general-purpose LLMs without specific text adaptation for visual tasks.

To address these challenges, we propose a novel approach to improve the vision-relevant abilities of text-based LLMs. We are motivated by the use of Braille codes by visually impaired readers, which allows them to understand the world despite not having access to optical system based perception. Our approach spans a variety of relevant skills, covering visual perception (classification), abstraction (summarizing), and reasoning capabilities (Q&A). Particularly, to improve perception related abilities of LLMs, we design a process for generating instruction-tuning data. This process utilizes large visual classification datasets and LLM generated *class descriptors*, which are then filtered for discriminative ability through feedback from a VLM. The filtered descriptors are used to fine-tune the LLM, improving its semantic knowledge for visual perception. To improve visual abstraction capabilities, we obtain supervision by pairing together detailed and short captions for images and videos. The LLM is trained to generate a short caption from the detailed one, which improves the LLM's ability to identify and focus on the most salient visual elements (hence, visual abstraction). For reasoning, we build our supervision using visual question answering datasets, however, as our goal is to train in the text domain, instead of visual input, information about the image or video is provided in the form of captions. The LLM is then trained to answer reasoning-based questions using the descriptions. The combination of all these three skills - perception, abstraction, and reasoning - together makes up our comprehensive IFT dataset, BRAILLEVISION-360K, designed to significantly enhance the vision-relevant capabilities of general-purpose LLMs.

We utilize BRAILLEVISION-360K to train a vision-specialized LLM, and experiment with different fine-tuning methods, including Supervised Fine-Tuning (SFT), Direct Preference Optimization

(DPO). We also propose a novel Fine-Grained SFT method, which assigns task-specific importance weights to tokens during the fine-tuning process and it outperforms SFT and DPO in this setting. This vision-specialized LLM outperforms generalized LLMs when utilized in a variety of vision tasks, such as assisting with visual classification and question answering using captions. We further train our instruction-tuned LLM in a LLaVA-like setting and observe superior performance on multi-modal benchmarks such as VQAv2, VizWiz, TallyQA etc.

In summary, our main contributions include:

- A method for automatically generating a diverse text-based instruction fine-tuning dataset, BRAILLEVISION-360K, capturing vision-centric skills for LLMs.
- An LLM, BRAILLEVISION-T, with specialized vision-related skills, which in collaboration with task specific modules like CLIP or a class agnostic detector, demonstrates improvement over off-the-shelf LLMs on tasks like image or video classification and open vocabulary object detection.
- A multi-modal LLM, BRAILLEVISION-V, demonstrating significant improvements in multi-modal QA tasks over standard MLLMs by leveraging our vision-specialized LLM as '*backbone*'.

## 2 RELATED WORK

**Instruction finetuning** emerged as a response to large language models producing outputs that fail to align with user intentions, even when scaled to significant sizes. This misalignment often results in outputs that are not beneficial to users. Researchers have explored various approaches to train models to follow instructions more effectively. One notable direction is linked to the concept of cross-task generalization in language models. This approach involves fine-tuning language models on a diverse set of publicly available natural language processing (NLP) datasets, typically prefaced with suitable instructions. The models are then evaluated on a separate group of NLP tasks that were not part of the training process. Sanh et. al Sanh et al. (2021) first applied this approach to LLM instruction tuning with 62 training datasets across 12 tasks, the concurrent Flan-V1 Wei et al. (2021) consists of 53 tasks, whereas Flan-v2 Chung et al. (2024) scales this paradigm up to 1836 tasks. The second popular source of instruction fine-tuning data is human feedback editing or ranking LLM responses. Another source of instruction tuning data is high-quality texts such as portions of academic textbooks Gunasekar et al. (2023) or specialized QA websites Yue et al. (2023b; 2024) consisting of text in a question/answer format. This technique has been successful in domain-specific LLMs targeted at math and science problems. Finally, for smaller and medium scale LLMs (e.g. 7B scale), larger teacher models like GPT-4 have also been used to create instruction tuning data Wang et al. (2023c); Geng et al. (2023); Chiang et al. (2023a); Taori et al. (2023a).

Instruction tuning aims to enhance LLMs' capacity to handle natural language questions. The underlying concept is that by employing supervised learning to teach a language model how to execute tasks outlined in instructions, it will develop the ability to follow directives, even for previously unseen tasks. General instruction tuning datasets such as FLAN Wei et al. (2021) and Vicuna Chiang et al. (2023a) focus on improving this instruction following capability broadly. It has been observed that such instruction tuning is not necessarily adequate for specialized domains, e.g. Vicuna finetuned models perform worse than base LLaMA models when used to create LLaVA-like multimodal LLMs Karamcheti et al. (2024). Some other specialized domains such as code generation and math solving have created specialized IFT datasets. In order to build our IFT dataset which is specialized towards improving performance on vision tasks, we first explore which capabilities are necessary and then select datasets to use to learn those capabilities.

**Domain specific LLMs** are most commonly seen in the Code and Math domains. Code-specific LLMs specialize at both the pre-training and instruction tuning stages. As large amounts of code are available from public open-source repositories on GitHub etc, pre-training specialized on code data is feasible. At the instruction tuning stage, code models are also trained to recover from errors, fix bugs, understand commit diffs, etc. Some popular code LLMs include WizardCoder Luo et al. (2023b), Code-LLaMA Roziere et al. (2023), Code-Qwen, CodeStral Mistral (2024) etc. Math is another domain where the weakness of generic LLMs has led to domain-specific instruction tuning. Approaches based on textbook data, procedural generation, and mining data from educational websites have found success at math tasks. WizardMath Luo et al. (2023a) and MaMMoTH Yue et al. (2023b) are some LLMs used for Math.

| Skill | Task | Dataset | | Size | |
|---|---|---|---|---|---|
| | | Image | Video | Image | Video |
| Perception | Classification by description | ImageNet21k | Kinetics400 | 112,210 | 4,000 |
| Abstraction | Summarization | FuseCap (COCO) | Ego4D | 113,287 | 44,000 |
| Reasoning | QA using descriptions | VQAv2 | ActivityNet (VCG) | 80,000 | 10,009 |

Table 1: Choice of tasks and corresponding datasets for each visual skill in BrailleVision-360k.

**LLMs have been increasingly applied to vision tasks**, particularly through integration with VLMs and multi-modal training. LLMs are combined with VLMs, in the zero-shot scenario, to understand and interpret visual inputs without requiring explicit training on the task. The VLM extracts meaningful features from the image, which are then translated into textual descriptions for the LLM to process, allowing to solve tasks such as image classification Menon & Vondrick (2023); Roth et al. (2023); Pratt et al. (2023), action recognition Lin et al. (2023), multi-modal open vocabulary detection Kaul et al. (2023), combining text captions across visual and audio modalities Chen et al. (2024), rewriting video subtitles Shvetsova et al. (2023), Visual anomaly detection Zhu et al. (2024), Hand-Object Interaction detection Lei et al. (2024) etc. This class of approach leverages the language model's extensive knowledge base to provide coherent responses, despite the absence of specific training data for that task. Furthermore, LLMs can be fine-tuned with visual tokens to build multi-modal LLMs, enabling them to process and understand visual inputs directly. This involves appending visual embeddings to the text input Liu et al. (2024), allowing the LLM to learn from both language and visual data simultaneously. By doing so, multi-modal LLMs can perform a wide range of vision tasks, such as image classification, object detection, and visual question answering.

However, all these approaches rely on generic LLMs which are not trained with any pre-training or instruction tuning data, specific to vision. In this work, we build a domain-specific instruction tuning dataset and use it to fine-tune LLMs to assist with vision tasks. Our model is a drop-in replacement for off-the-shelf LLMs utilized in prior works and provides a significant improvement due to its domain-specific vision centric fine-tuning.

## 3 CONSTRUCTING THE BRAILLEVISION-360K DATASET

In this section we discuss the creation of our IFT dataset, BRAILLEVISION-360K, focusing on the purpose of each component skill, the chosen task and the source datasets. In order to unlock the vision relevant capabilities of LLMs, we first discuss which capabilities and datasets to use to build our IFT training mix. We decide to focus on vision capabilities in three broad areas: perception, abstraction, and reasoning. These skills together cover the vast majority of computer vision tasks of interest. In perception, we focus on the simplest semantic level: classification, the ability to identify specific objects and actions and semantically describe and relate them to other concepts. With regards to abstraction, we focus on the capability of summarization: the ability to distill down a long visual description of an image or video into a short sentence containing the most salient details. Finally, with reasoning, we focus on question answering based on visual inputs, which tests the ability to utilize perceived information to draw logical conclusions, make inferences, and generate accurate responses grounded in the visual context provided. We carefully create instruction fine-tuning data for each of our vision-relevant skills. For each skill, we have a separate source of data for image and video tasks. We provide dataset overview and stats in Table 1 while a few samples of each component of our dataset are provided in Figure 5 of our Appendix.

### 3.1 PERCEPTION

Perception is how an agent acquires information about the current state of the world to update its world model. In the mammalian visual system, perception relies on the optic nerve to gather input and the visual cortex to interpret it. The feature-integration Treisman & Gelade (1980) theory of attention proposes that when multiple distinct features are required to identify or differentiate objects in a display, attention must be focused on each stimulus individually in a sequential manner. The goal of the perception component of our skills training is to enhance the LLM's ability to generate concept attributes or class descriptors sequentially.

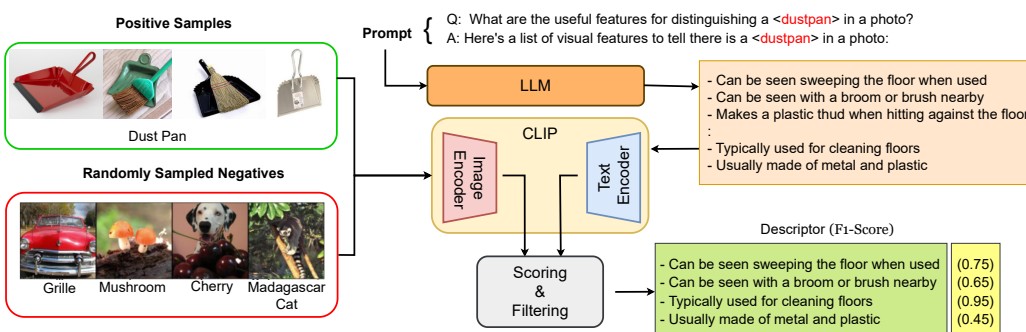

Figure 2: Creating IFT dataset for learning perception skills using CLIP Feedback. Each class descriptor generated by the LLM is scored by CLIP for its effectiveness in visually distinguishing the target class from a random sample of other classes.

**Classification:** For images, we probe the perception abilities of the LLM by prompting it to generate class descriptors for different classes and then employ these descriptors for zero-shot classification of images using CLIP. This approach is broadly similar to the one proposed in Classification-by-Description Menon & Vondrick (2023). However, unlike that work, we also investigate the utility of each individual descriptor generated by the LLM, filtering out less useful ones to ensure the downstream LLM trained on this dataset generates only useful visual descriptions. For videos, we follow a framework similar to that of the images. We utilize Kinetics-400 Kay et al. (2017) for videos and ImageNet-21K Deng et al. (2009); Ridnik et al. (2021) for images. Next, we elaborate on our visual descriptor filtering strategy utilizing CLIP feedback.

**Scoring Class Descriptors using CLIP Feedback:** The process of creating an Instruction Following Tuning (IFT) dataset for teaching a language model to generate visually discriminative class attributes involves several steps. Consider a large labeled dataset of images/videos, such as ImageNet21k. Images in the dataset belong to a set of image classes $C = \{C_0, C_1, \ldots, C_n\}$ and the assignment of labels to samples is represented by $D = \{(x_1, l_1), (x_2, l_2), \ldots, (x_m, l_m)\}$, where each pair $(x_i, l_i)$ represents an image and its corresponding label, with $l_i \in C$. The process begins by randomly selecting a class $C_k$ and sampling two sets of images: $X_{\text{neg}} = \{x_j \mid (x_j, l_j) \in D, l_j \neq C_k, |X_{\text{neg}}| = N\}$, containing $N$ negative samples from classes other than $C_k$, and $X_{\text{pos}} = \{x_p \mid (x_p, l_p) \in D, l_p = C_k, |X_{\text{pos}}| = N\}$, comprising $N$ positive samples from class $C_k$. Next, a Large Language Model $L$ is utilized to generate a set of class descriptors $\{d_{k,0}, d_{k,1}, \ldots, d_{k,q}\} = L(C_k, \text{prompt})$ based on the class $C_k$ and a prompt template. Once generated, these descriptors need to be scored for their usefulness for the classification task. Hence, each descriptor along with the class name, are then passed through a CLIP text encoder $T_{\text{CLIP}}$, resulting in encoded representations $t_{k,j} = T_{\text{CLIP}}(C_k \oplus d_{k,j})$, where $\oplus$ operator represents a rule based operation for combining descriptor and class name. Pseudo-code for this operation is provided in the Appendix (Algorithm 1). Simultaneously, the sampled images (both the negatives, and positives) are processed through a CLIP image encoder $V_{\text{CLIP}}$, producing visual embeddings $v_x = V_{\text{CLIP}}(x)$ for each image $x$. Finally, for each descriptor $d_{k,j}$, an F1 classification score is calculated: $f(C_k, d_{k,j}) = F1\_score(\{sim(v_x, t_{k,j}) \mid x \in X_{\text{neg}} \cup X_{\text{pos}}\})$, based on the cosine similarity (denoted as $sim(.)$) of the descriptors which measures the effectiveness of each descriptor in classifying the images. Figure 2 illustrates this process.

## 3.2 ABSTRACTION

Abstraction is the cognitive process of simplifying complex information by distilling it into its most essential, general, or fundamental elements. It involves focusing on the relevant or important features of an object, idea, or concept while ignoring the less significant details. This streamlined organization helps prevent memory overload and expand processing capabilities, thus improving both retention and problem-solving abilities Rogers (2024). Many theories of human visual recognition, such as, recognition by components Biederman (1987) posit that the human visual system involves a significant degree of abstraction, i.e. visual scenes are recognized by 'summarizing' them into a set of components. It has also been suggested that understanding human text involves the implicit construction of summaries Graesser et al. (1994). Prior work in education research has demonstrated

that students can be taught essential skills through learning to summarizing Boujaoude (1992); Wittrock & Alesandrini (1990). In addition to educational applications, these principles are also relevant in fields like natural language processing (NLP) and computer vision. In the NLP domain, Google's Muffin Wei et al. (2022) and Flan Chung et al. (2024) instruction tuning datasets include text summarization tasks, focusing on news, dialogue, and documents. While summarization is commonly associated with text, similar principles can be applied in visual tasks to condense information. Prior works have demonstrated that visual abstraction is useful in tasks like Image Segmentation Shimoda & Yanai (2016), Video event detection Gan et al. (2015), and Video Question Answering Yu et al. (2024); Zhang et al. (2023).

**Summarization:** We pick the task of visual description summarization to teach the model abstraction skills most relevant to computer vision. Specifically, the model is provided with a highly detailed description of an image or video and has to generate a short description that still describes the sample adequately. This task requires the model to focus on the key salient details while discarding irrelevant ones. In particular, for images, we use image captions for the COCO dataset Lin et al. (2014) for this part of our dataset. The detailed captions that need to be summarized are obtained from the FUSECAP Rotstein et al. (2024) paper, whereas the short captions are obtained from the original COCO dataset. As both FUSECAP and COCO provide multiple captions per image, we choose the best one on the basis of its BLIPScore, choosing the caption with the highest similarity to the image. For video, we use the narrations from the Ego-4D Grauman et al. (2022) dataset, which are typically provided every for second as input. The output summaries cover each 5-minute chunk of video with a single short caption.

### 3.3 REASONING

Reasoning plays a crucial role in intelligence by connecting perception and abstraction to actionable insights and decisions. Simply building a maximally accurate (perfect perception) and parsimonious (perfect abstraction) representation of input information is insufficient to achieve true cognition. For a system to achieve cognition requires *reasoning*, the ability to process information and modify its behavior in response Milkowski (2013). Reasoning encompasses a broad range of abilities such as reasoning by analogy to generalize to novel situations Gentner & Hoyos (2017), recovering the physical state of the world (e.g. a 3D model) from limited information (e.g. a 2d image) Wu et al. (2017), inferring causal relationships from sparse data Gopnik et al. (2004), etc. Visual reasoning tasks, in particular, require the system to examine complex visual stimuli, encompassing both foreground subjects, contextual background information, etc. and answer questions based on them.

**Question Answering:** We design a text-based version of the Visual Question Answering (VQA) task, where a large language model (LLM) answers questions about images and videos based on detailed textual descriptions instead of the visual content itself. For the image-based tasks, we use the VQAv2 dataset Antol et al. (2015), which contains images from the COCO (Common Objects in Context) dataset. However, instead of relying on COCO's original captions, we use captions generated by the FUSECAP model, which provides more detailed and informative descriptions. These captions help the LLM understand the image content to answer the questions.

For video-based tasks, we use the ActivityNet dataset Caba Heilbron et al. (2015), which includes videos depicting various human activities, accompanied by captions. The questions for the videos are taken from the VideoChatGPT Maaz et al. (2024) dataset, which consists of more complex and challenging questions requiring an understanding of temporal dynamics. Unlike static images, answering video questions involves grasping the sequence and flow of events over time, making the task more challenging. The model must understand not only individual frames but also the relationship between events over time. In both cases, the LLM is trained to interpret these detailed text descriptions to answer questions typically posed in visual tasks. However, for videos, the challenge lies in understanding the temporal structure and dynamics, which adds another layer of complexity compared to image-based tasks.

## 4 TRAINING VISION SPECIALIZED LLMS

After building the dataset, we focus on fine-tuning the base LLM to develop BRAILLEVISION-T, our vision specialized text LLM. We experiment with SFT and DPO, existing methods for LLM training,

and also test our proposed Fine-Grained (Fine SFT) method. Finally, we fine-tune a LLaVA-Like model (named BRAILLEVISION-V) with our specialized LLM as the '*backbone*' to solve various multi-modal tasks.

## 4.1 TRAINING BRAILLEVISION-T

Once the dataset is created, our next step is to use it to train the LLM. Following the literature, we evaluated Supervised Fine-Tuning (SFT, also known as behavior cloning in the Reinforcement Learning literature) and Direct Preference Optimization (DPO) and also proposed our own Fine-Grained SFT (FineSFT) method for training the LLM.

In supervised fine-tuning, the model is simply trained using the language-modeling loss (i.e. next token prediction) over the target dataset. Every single token in the SFT has the same weight, which can sometimes lead to undesirable overfitting and hurt generalization.

If $\pi_\theta$ represents the model, the SFT loss over a set of prompts $\mathbf{x}$ and expected outputs $\mathbf{y}$ is given by:

$$\mathcal{L}_{\text{SFT}}(\pi_\theta, \mathbf{x}, \mathbf{y}) = -\mathbb{E}_{\mathbf{x} \sim q(\cdot), \mathbf{y} \sim p_{\text{data}}(\cdot|\mathbf{x})} \left[ \log \pi_\theta(\mathbf{y}|\mathbf{x}) \right].$$

Direct Preference Optimization, on the other hand, is a method for sequence-level supervision, where the product of the relative log probabilities (relative to the reference model) of a desired or chosen output is raised compared to the product of the relative log probabilities of an undesired (rejected) output. DPO has been utilized in state-of-the-art open-source LLMs and outperforms SFT. If $y_w$ and $y_l$ represent the chosen and rejected responses, the DPO loss is given by:

$$\mathcal{L}_{\text{DPO}}(\pi_\theta; \pi_{\text{ref}}; \mathbf{x}, \mathbf{y}) = -\mathbb{E}_{(\mathbf{x}, y_w, y_l) \sim \mathcal{D}} \left[ \log \sigma \left( \beta \log \frac{\pi_\theta(y_w|\mathbf{x})}{\pi_{\text{ref}}(y_w|\mathbf{x})} - \beta \log \frac{\pi_\theta(y_l|\mathbf{x})}{\pi_{\text{ref}}(y_l|\mathbf{x})} \right) \right].$$

.

However, DPO is more complex than SFT and requires keeping both the reference model and the model undergoing finetuning in memory. DPO can also result in the LLM unlearning base knowledge Yan et al. (2024) due to excessive lowering of probability for rejected response. Hence, we introduce a fine-grained SFT approach where we weigh the loss at each token. The weights can be task-specific; for instance, for the classification task, the F1 score for the corresponding descriptor from CLIP can be used, while for summarization, BLIPScore can be used as the token weights. This provides us with more fine-grained supervision than SFT while avoiding the problems associated with DPO. Our loss requires the token loss weights as an additional input, $\mathbf{w}$. Fine-SFT is illustrated in Figure 3 and its loss equation is given by:

$$\mathcal{L}_{\text{SFT}}^{\text{weighted}}(\pi_\theta; \mathbf{x}, \mathbf{y}, \mathbf{w}) = -\mathbb{E}_{\mathbf{x} \sim q(\cdot), \mathbf{y} \sim p_{\text{data}}(\cdot|\mathbf{x})} \left[ \sum_{i=1}^{n} w_i \log \pi_\theta(y_i|\mathbf{x}) \right].$$

## 4.2 TRAINING MULTI-MODAL BRAILLEVISION-V

The logical progression for our experimental framework is to integrate our LLM into a comprehensive multi-modal LLM training architecture. Specifically, we have chosen to utilize the LLaVA (Large Language and Vision Assistant) architecture Liu et al. (2024). In our experiments, we closely adhere to the one stage training methodology outlined by Prismatic-VLMs Karamcheti et al. (2024). This involves the alignment of a SigLIP Zhai et al. (2023a) vision encoder with a 7 billion parameter LLM, by training on the LLaVA-Instruct-v1.5 dataset.

Our goal with this experiment is to assess the impact of our text-based instruction tuning on subsequent performance across various multi-modal benchmarks. The critical distinction between our Multi-modal Large Language Model (MLLM) and the established baseline lies in the text instruction tuning phase of the process. For comparison, the baseline methodology presents results from two scenarios: one without any text instruction tuning, and another utilizing the `Vicuna` text instruction tuning approach. Our experimental design aims to evaluate and contrast the effectiveness of these two baseline approaches against our novel text instruction tuning method.

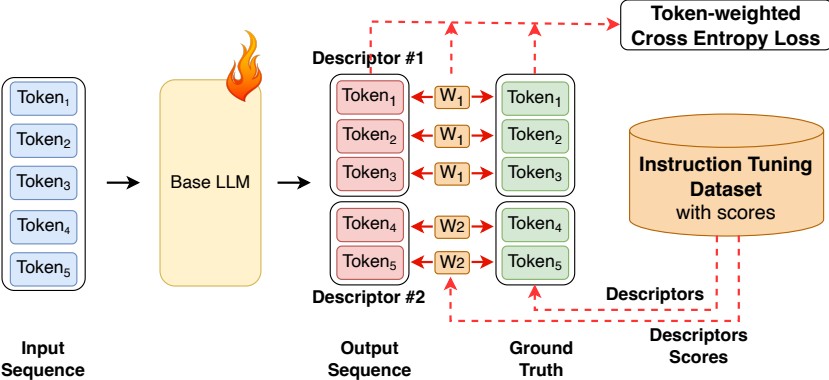

Figure 3: Finetuning an LLM can be performed through simple Supervised Fine Tuning (SFT) where each token gets equal weight. Our proposed FineSFT method on the other hand weights tokens during training based on the discriminativeness score of their corresponding visual descriptors .

| Method | Image Classification | | | | | Video Classification | |
|---|---|---|---|---|---|---|---|
| | **CalTech** | **Pets** | **Cars** | **Food** | **SUN** | **UCF-101** | **HMDB-51** |
| (a) CLIP (C) | 93.3 | 88.2 | 65.6 | 85.3 | 62.6 | 64.5 | 37.5 |
| (b) C + M-7B-Instruct | 94.5 | 89.6 | 75.5 | 87.5 | 69.4 | 68.6 | 46.1 |
| **Ours (Visual class descriptor instruction tuned)** | | | | | | | |
| (c) C+M-7B (SFT) | 94.9 | 92.5 | 78.3 | 88.2 | 72.3 | 75.8 | 49.5 |
| (d) C+M-7B (DPO) | 95.3 | 92.7 | 78.7 | 88.8 | 73.5 | 77.2 | 51.7 |
| (e) C+M-7B (Fine-SFT) | 95.7 | 93.1 | 79.1 | 90.4 | 73.9 | 78.1 | 52.6 |
| **Our Gains** ((e) - (b)) → | ↑ **1.2** | ↑ **3.5** | ↑ **3.6** | ↑ **2.9** | ↑ **4.5** | ↑ **9.5** | ↑ **6.5** |

Table 2: LLM assisted CLIP zero-shot image and video classification (following Menon & Vondrick (2023)). Our visual classification instruction-tuned LLM beat the off-the-shelf LLM by 3% (image) and 8% (video) on average. M-7B → Mistral-7B.

## 5 EXPERIMENTS AND RESULTS

In this section, we present our experimental results. First, we focus on evaluating the ability of our approach to improve the ability of LLM to assist classification tasks. Next, we evaluate the Image-QA skills of a LLaVA-like model trained using our LLM. The corresponding ability for Video-QA tasks utilizes the LLM as both summarizer and reasoner, demonstrating its impact on both skills. Finally, we carry out ablations to assess the impact of each set of skills in our IFT dataset.

### 5.1 LLM ASSISTING VISION MODELS

**Classification by Description:** Our initial experiments focus on the task of classification by description using our finetuned LLMs to assist a CLIP model. Our models are finetuned using supervision from the ImageNet-21k (image) and Kinetics-400 (video) datasets, and tested on a benchmark of Image and Video Classification datasets including CalTech-101, Oxford-IIIT Pets-37, Stanford Cars-196, Food-101 (image) and UCF-101 & HMDB-51. The detailed results are presented in Table 2. These results demonstrate that finetuning the LLM on vision specific text data can improve its zero-shot classification abilities, by 3% (image classification) and 8% (video classification) on average. We also ablate three different methods of instruction tuning the LLM: SFT (row-*c*), DPO (row-*d*) and our proposed Fine-SFT (row-*e*). Fine-SFT outperforms the others, due to providing fine-grained token level weighted supervision.

**Zero-Shot Cross-Task Transfer:** We further test our LLM on a perception task not seen during training to evaluate if our IFT transfers across tasks. For this we pick the task of few shot open vocabulary object detection, following the scheme from prior work MMC-OVOD Kaul et al. (2023).

| Method | LVIS-Base $\rightarrow$ LVIS | | LVIS-Base + IN-L $\rightarrow$ LVIS | |
| --- | --- | --- | --- | --- |
| | $AP_r$ (%) | mAP (%) | $AP_r$ (%) | mAP (%) |
| DETIC | - | - | 24.6 | 32.4 |
| MMC-OVOD | 19.3 | 30.6 | 27.3 | 33.1 |
| MMC-OVOD (with our LLM) | 21.5 ↑ **2.2** | 31.7 ↑ **1.1** | 29.4 ↑ **2.1** | 34.7 ↑ **1.6** |

Table 3: Open Vocabulary Object Detection results. $AP_r \rightarrow$ AP for Rare classes.

In this approach an LLM is prompted to provide visual descriptions of the category of interest, for which embeddings are generated using CLIP text encoder, and fused with CLIP image embeddings for the few shot exemplars to create a classifier. Outputs from a class agnostic object detector (CenterNet2 Zhou et al. (2021) with ResNet50 He et al. (2016) backbone) can then be classified among any given vocabulary of classes using the aforementioned classifier. We replace the LLM component of the system with our BrailleVision finetuned LLM and observe gains in object detection performance on LVIS dataset, both with or without using ImageNet21K-LVIS overlap set as additional image level data (Fig. 3). Performance on rare classes in particular rose by more than 2%, which is a significant improvement on this hard task. These results establish that our vision-centric IFT improves the perception ability of LLMs in general, even beyond tasks it is trained on.

| LLM Summarizer | Q/A | EgoSchema-Val Top-1 Acc. | NeXT-QA | ActivityNet-QA |
| --- | --- | --- | --- | --- |
| LLaMA2 | LLaMA2 | 34.0 | 50.1 | 50.8 |
| Vicuna | Vicuna | 34.4 | 50.7 | 51.3 |
| **Ours** | **Ours** | **41.7** ↑ **7.3** | **58.2** ↑ **7.5** | **55.6** ↑ **4.3** |

Table 4: Video-Question Answering following the LLoVI framework.

**Video QA:** We follow LLoVI Zhang et al. (2024) framework to evaluate our model on Video Question Answering. This framework typically consists of three stages: captioning, caption summarization and question answering. Captions are generated using either an expert captioner (LaViLa for EgoSchema) or an MLLM (LLaVA-1.5 for NeXT-QA and ActivityNet), which is common across methods, Summarization and QA are done by a standard LLM. Results in Table 4 show that our LLM outperforms generic baselines in both the Summarization as well as QA part of the benchmark.

| Model | Text IFT | In-Domain | | Zero-Shot | | |
| --- | --- | --- | --- | --- | --- | --- |
| | | VQAv2 | GQA | VSR | VizWiz | TallyQA |
| **Prismatic** | None | 77.08 | 62.44 | 63.67 | 55.98 | 59.22 |
| **LLaVA-1.5** | Vicuna | 77.09 ↑ **0.0** | 62.57 ↑ **0.1** | 51.47 ↓ **12.2** | 54.33 ↓ **1.7** | 61.63 ↑ **2.4** |
| **Ours** | BrailleVision | 78.32 ↑ **1.2** | 63.49 ↑ **0.9** | 63.91 ↑ **0.2** | 57.15 ↑ **1.2** | 61.75 ↑ **2.5** |

Table 5: Benchmark evaluation of MLLM trained using our BrailleVision-360k instruction tuned LLM as 'backbone' outperforms Vicuna (LLaVA) and Base LLaMA2 (Prismatic VLM) LLMs.

## 5.2 MULTI-MODAL LLM

The next direction we investigate is using our vision specialized LLM for MLLM training utilizing a LLaVA-1.5-like framework, specifically, the Prismatic VLLM Karamcheti et al. (2024) setting. Prior work had demonstrated that general text instruction tuning as done in Vicuna does not improve MLLM's performance on mutlti-modal tasks. We test our MLLM on a variety of Image QA tasks and then on specific benchmarks that focus on hallucinations.

**Image QA:** The results in Table 5 show that using our LLM to train an MLLM outperforms both using a base LLaMA-2 model and a Vicuna instruction finetuned model. As previous research Karamcheti et al. (2024) has indicated that general instruction tuning does not significantly benefit the adaptation of Large Language Models (LLMs) to multi-modal contexts, our finding demonstrates that text instruction tuning can be useful, but only if its focused on vision relevant skills.

**Effect on Hallucinations:** A key limitation of multi-modal LLMs is their propensity to hallucinate which they inherit from their pre-trained LLM '*backbone*'. We evaluate our MLLM for object

hallucination using POPE Li et al. (2023b). HallusionBench Guan et al. (2024) on the other hand tries to detect when conclusions are made by the model ignoring visual input due to strong language prior and when visual inputs are misinterpreted, resulting in overly confident but incorrect statements by the model. Our IFT results in fewer hallucinations (See Tab. 6), particularly HallusionBench, which is focused on hallucinations caused by an incorrect language prior.

| Model | Text IFT | POPE-Overall | POPE-Adversarial | HallusionBench |
|---|---|---|---|---|
| **Random Baseline** | - | 50.0 | 50.0 | 45.96 |
| **Prismatic** | None | 86.74 | 84.5 | 46.06 |
| **LLaVA-1.5** | Vicuna | 86.57 ↓ **0.2** | 84.0 ↓ **0.5** | 46.06 |
| **Ours** | BrailleVision | 87.21 ↑ **0.5** | 86.1 ↑ **1.6** | 48.71 ↑ **2.6** |

Table 6: Effect of our Text IFT on propensity to hallucinate.

| IFT Training Split | | | Classification | | VQA | |
|---|---|---|---|---|---|---|
| Classification | Summarization | Reasoning | Image | Video | Image | Video |
| ✔ | ✔ | ✔ | **86.4** | **78.1** | **78.3** | **41.7** |
| ✔ | ✔ | - | 86.1 | 78.0 | 76.5 | 34.8 |
| ✔ | - | ✔ | 85.4 | 78.3 | 78.1 | 39.5 |
| - | ✔ | ✔ | 80.7 | 65.6 | 77.0 | 41.5 |

Table 7: Skills Dataset Ablations for BRAILLEVISION-360K.

## 5.3 ABLATIONS

**Dataset Ablation:** We ablate different components of our instruction tuning dataset in Table 7. Particularly, from 2nd and 3rd rows, we observe VQA performance degrades if we remove summarization and reasoning related data from BRAILLEVISION-360K. Similarly, removing visual perception related classification data significantly drops image and video classification performance. Overall, these results show that each component of our dataset is necessary for all around improvement.

**Vision Feedback Model Ablation:** We also ablate the vision-language feedback model, replacing CLIP successively with MAWS Singh et al. (2023), which is a MAE-like model later aligned with language modality and Visually Enriched-CLIP Lai et al. (2024), which is trained with longer, detailed synthetic captions. We find that MAWS underperforms CLIP while VE-CLIP performs similarly. Detailed results are presented in Table 8.

**LLM Ablation:** We experiment with different base LLMs for finetuning with BRAILLEVISION-360K. We find that Mistral Jiang et al. (2023) and LLaMA2 Touvron et al. (2023) both perform similarly (Table 9), with LLaMA outperforming at VQA slightly, and Mistral better at classification.

| Vision Feedback | Classification | | VQA | |
|---|---|---|---|---|
| | Image | Video | Image | Video |
| CLIP | 86.4 | **78.1** | **78.3** | 41.7 |
| MAWS | 85.8 | 76.5 | 77.4 | 41.0 |
| VE-CLIP | **86.5** | 78.0 | 78.1 | **41.7** |

Table 8: Vision Feedback Ablation.

| LLM | Classification | | VQA | |
|---|---|---|---|---|
| | Image | Video | Image | Video |
| Mistral-7B | **86.4** | **78.1** | 78.3 | 41.7 |
| LLaMA2-7B | 86.3 | 77.4 | **78.5** | **42.8** |

Table 9: Base LLM Ablation.

## 6 CONCLUSION

In this paper, we introduced BRAILLEVISION, a novel approach to enhance the visual capabilities of Large Language Models through vision-specific instruction fine-tuning. By focusing on key vision-related skills—perception, abstraction, and reasoning—we demonstrated how targeted text-based training can significantly improve an LLM's performance across a range of visual tasks. Our results showed that this specialized instruction tuning, leads to better performance in visual classification, and zero shot task transfer to other perception tasks such as open vocabulary detection. Additionally, by integrating our fine-tuned LLM into a multimodal large language model, we observed notable improvements in multi-modal tasks, such as image and video question answering, and reduced hallucinations. This work underscores the importance of aligning LLM training with domain-specific tasks, showing that specialized fine-tuning can significantly boost multimodal intelligence.

ETHICS STATEMENT

Our method does not involve use of any human subjects and the authors do not have any conflicts of interest to declare. As our method is based on the use of pre-trained open weight LLMs, it could potentially be impacted by fairness and bias concerns inherited from the base LLM, however we believe our method does not introduce any new fairness issues.

REPRODUCIBILITY STATEMENT

Our experiments are carried out with open weight models and public domain datasets and can be reproduced by closely following the steps outlined in the paper. We will also release the code and trained weights upon acceptance of the paper.

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

# APPENDIX

## A  SENTENCIFY ALGORITHM

The algorithm used for converting descriptors generated by the LLM into sentences used for prompting CLIP is explained in Alg. 1. This follows the approach from Menon & Vondrick (2023) and handles common cases to ensure meaningful sentences are generated.

---

**Algorithm 1** Function to sentencify a descriptor to prompt CLIP

---

1: **function** SENTENCIFY_DESCRIPTOR(descriptor, classname)
2:  **if** descriptor starts with "a" OR descriptor starts with "an" **then**  ▷ Handles descriptors introducing a noun or noun phrase
3:    **return** "a photo of a " + classname + " which is " + descriptor
4:  **else if** descriptor starts with "has" OR "often" OR "typically" OR "may" OR "can" **then** ▷ Handles verb phrases describing characteristics or abilities
5:    **return** "a photo of a " + classname + " which " + descriptor
6:  **else if** descriptor starts with "used" **then**  ▷ Handles descriptors describing purpose or function
7:    **return** "a photo of a " + classname + " which is " + descriptor
8:  **else**                ▷ Handles features or qualities that something has
9:    **return** "a photo of a " + classname + " which has " + descriptor
10:  **end if**
11: **end function**

---

## B  CREATION OF IFT DATASET FOR SUMMARIZATION

The process as discussed in Section 3.2 is illustrated in Fig. 4.

## C  COMPARISON TO OTHER TEXT IFT DATASETS

In Table 10 we compare our IFT dataset against prior works, our dataset is comparable in scale to the largest IFT datasets, while covering unique capabilities and created through a novel process (use of CLIP feedback).

# D SAMPLES FROM BRAILLEVISION-360K

We provide samples from each task in our dataset in Fig. 5.

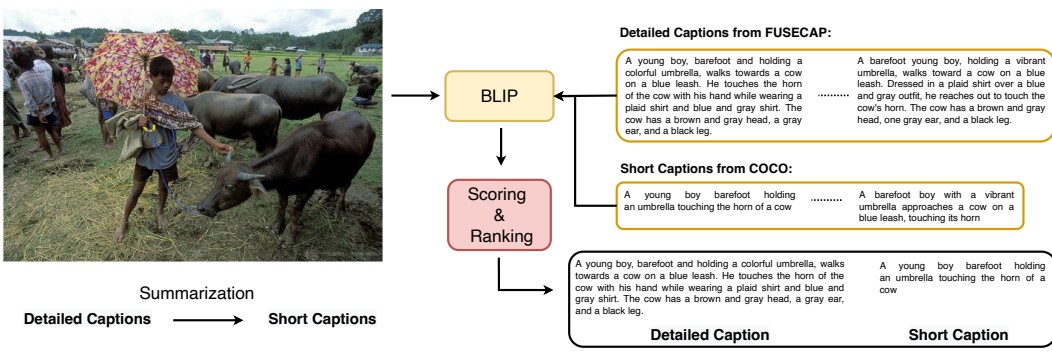

Figure 4: Creation of IFT dataset for Summarization Task

| Dataset | Domain | Size | Generation Process |
|---|---|---|---|
| SuperNI Wang et al. (2022b) | Generic | 96.9K | NLP Datasets + Hand Written Prompts |
| Flan V2 Longpre et al. (2023) | Generic | 100K | NLP Datasets + Hand Written Prompts |
| Dolly Conover et al. (2023) | Generic | 15.1K | Hand Written |
| Open Assistant 1 Köpf et al. (2024) | Generic | 34.7K | Hand Written |
| Self Instruct Wang et al. (2022a) | Generic | 82.4K | GPT-3 |
| Unnatural Instructions Honovich et al. (2022) | Generic | 68.4K | GPT3 (davinci-002) |
| Alpaca Taori et al. (2023b) | Generic | 52K | GPT3 (davinci-003) |
| GPT4-Alpaca Peng et al. (2023) | Generic | 52K | GPT-4 |
| Baize Xu et al. (2023) | Generic | 210K | ChatGPT |
| ShareGPT Chiang et al. (2023b) | Generic | 168.8K | ChatGPT |
| Code-Alpaca Chaudhary (2023) | Coding | 20K | GPT3 (davinci-003) |
| CodeContest | Coding | 13.6K | Programming Contests |
| CommitPackFT Muennighoff et al. (2023) | Coding | 702K | GitHub Commits |
| ChatDoctor Li et al. (2023c) | Medical | 115K | - |
| DISC-Med-SFT Bao et al. (2023) | Medical | 464K | - |
| DISC-Law-SFT Yue et al. (2023a) | Law | 403K | - |
| Lawyer LLaMA SFT Huang et al. (2023) | Law | 21K | - |
| Lila Mishra et al. (2022) | Math | 272K | - |
| MathInstruct Yue et al. (2023b) | Math | 262K | GPT-4 |
| MetaMathQA Yu et al. (2023) | Math | 395K | GPT-3.5 |
| MathCodeInstruct Wang et al. (2023a) | Math | 80K | GPT-4 |
| WizardMath Luo et al. (2023a) | Math | 96K | GPT-3.5 |
| ToRA Gou et al. (2023) | Math | 16K | GPT-4 |
| OpenMathInstruct-1 Toshniwal et al. (2024) | Math | 1.8M | Mixtral |
| Ours | Vision | 360K | Vision Datasets + Mixtral + CLIP Feedback |

Table 10: Comparison of text instruction tuning datasets with their size and generation process.

| Tasks | |
|---|---|
| **Image Classification** | |
| **Prompt** | What are the visual features that distinguish a <class label> in a photo? |
| **Input** | Class Label: Volcano |
| **Output** | • A large cone shaped mountain • A crater at the top of the mountain 
 • Lava and Ash flowing from the crater • A plume of smoke and ash rising |
| **Video Classification** | |
| **Prompt** | Return a numbered list of just the verb phrase for the specific action steps a person would perform while doing <class label> |
| **Input** | Class Label: Archery |
| **Output** | • Stand in a stable position • Nock the arrow to the bowstring • Draw the bowstring back 
 • Aim at the target • Release the bowstring • Follow through with the shot 
 • Retrieve the arrow from the target |
| **Image Summarization** | |
| **Prompt** | Below is a detailed description of an image. Summarize the main points of this text. |
| **Input** | A kitchen with a white cabinet, silver stain-less steel sink, counter top, brown floor and closed white doors. The ceiling is ...... black oven, silver pot, and a white microwave are visible. |
| **Output** | Galley kitchen with cabinets and appliances on the sides. |
| **Video Summarization** | |
| **Prompt** | Below are the detailed captions of the frames from a video. Summarize the main points of this text. |
| **Input** | Person A sits in the room. Person B sits in the room. Person C hands playing cards to B. |
| **Output** | #Summary C was in the room and played cards with person A |
| **Reasoning - Image** | |
| **Prompt** | Please provide a single-letter answer (A, B, C, D, E) to the following multiple-choice question. Your answer must be one of the letters (A, B, C, D, or E) and no other response or explanation is allowed. You are given some language descriptions of an image. Here are the descriptions: <Narration>. You are going to answer a multiple-choice question based on the descriptions, and your answer should be a single letter chosen from the choices. Here is the question: <Question>. Here are the choices: A: <A> B: . |
| **Input** | Narration: Crispy, chewy crust with rich tomato sauce. Gooey melted mozzarella cheese. Thin slices of spiced salami (pepperoni). Crispy edges on the pepperoni when baked. Savory and slightly spicy flavor profile. 
 Question: Is this a vegetarian pizza? 
 <A>: Yes : No |
| **Output** |  |
| **Reasoning - Video** | |
| **Prompt** | Please provide a single-letter answer (A, B, C, D, E) to the following multiple-choice question. Your answer must be one of the letters (A, B, C, D, or E) and no other response or explanation is allowed. You are given some language descriptions of a first-person view video. Here are the descriptions: <Narration>. You are going to answer a multiple-choice question based on the descriptions, and your answer should be a single letter chosen from the choices. Here is the question: <Question>. Here are the choices: A: <A> B:  C: <C> D: <D>. |
| **Input** | Narration: Approach the runway. Run down the runway. Plant the pole in the box. Jump off the ground ... 
 Question: What happened before Pole Vault? 
 <A>: Running : Jumping <C>: Falling <D>: Celebrating |
| **Output** | <A> |

Figure 5: Samples from our text instruction fine-tuning dataset for unlocking perception, summarization, and reasoning capabilities, with a video and image counterpart respectively.

