# OpenReview forum: "BrailleVision: Text Instruction Tuning of LLMs to Improve Visual Skills"
_ICLR.cc/2025/Conference — Submitted to ICLR 2025_

### Official Review · Reviewer_ZP38 · 2024-10-30

**Soundness:** 2
**Presentation:** 2
**Contribution:** 2
**Rating:** 3
**Confidence:** 4

**Summary:**

This paper focus on improving the visual reasoning ability of text-based LLM on VL tasks, and propose a new dataset called BrailleVision-360k covering the scopes of visual perception, abstraction and spatiotemporal reasoning. A new Fine-SFT tuning approach is also proposed for text-based LLM. However, the study problem receive limited attention in recent MLLM study, and the authors lack enough proofs to highlight the significance of this task, limiting its potential contributions.

**Strengths:**

The propose of a new dataset called BrailleVision to teach text-based LLMs visual skills, such as visual perception, abstraction, and spatial temporal reasoning. The experiments shows the effectiveness of this dataset for text-based LLMs.

**Weaknesses:**

1. The importance of the studied problem in this paper is questioned, i.e., improving the visual ability of only text-based LLM. As described in the introduction, I think the most popular paradigm of MLLM is the first one, i.e., extending LLM to vision-language task, which is adopted by most existing MLLMs. In contrast, the mentioned second paradigm seems receiving  much less attention in both academia and industry. For instance, the papers cited by the authors in the introduction are before 2024, and only one is published in 2023. I would suggest the authors to give more proofs to indicate the importance of the studied problem, otherwise, the contribution will be very limited.

2. The experimental section is not sufficient. If the authors think that text-based LLM is an optimal solution for multimodal tasks, more comprehensive comparisons are required. In particular, text-based LLM for VL tasks also requires an VLM as a supplement, thus its overall parameter-scale is in fact similar with existing end-to-end MLLMs. So more comparisons are needed, for instances, the comparisons with more advanced MLLMs on more MLLM benchmarks.

Minors:

1. Under the task background and study problem of this paper, the description of ``Current multimodal large language models (MLLMs)
incorporate general-purpose LLMs through multimodal instruction tuning. These LLMs, however, lack prior vision centric text based training, potentially limiting their effectiveness'' seems not very suitable. At the first glimpse, I thought this paper is to study the VL instruction tuning for common MLLMs.

**Questions:**

Most of my concerns and questions are given in the weakness part. In fact, I think that the proposed BrailleVision dataset still has great potential values if this dataset can be extended to a multimodal one for the VL instruction tuning of common MLLMs. So is it possible to extend this dataset for the common VL instruction tuning of MLLMs, and what benefits it can get?

---

### Official Review · Reviewer_ac2k · 2024-10-31

**Soundness:** 2
**Presentation:** 1
**Contribution:** 2
**Rating:** 3
**Confidence:** 4

**Summary:**

This paper proposes BrailleVision, a method to enhance the vision-related capabilities of large language models (LLMs) through instruction fine-tuning with vision-centric text data. The authors construct an instruction-tuning dataset designed to teach skills such as visual perception, abstraction, and spatio-temporal reasoning without the use of visual data, analogous to how Braille codes are utilized by the visually impaired.

Experimental results demonstrate that the proposed vision-specialized LLM achieves significant performance gains in tasks such as visual classification, open vocabulary detection, and visual question answering (VQA).

**Strengths:**

1. The concept of teaching visual skills to LLMs without relying on visual data is intriguing. I appreciate the motivation drawn from Braille codes, which enable visually impaired individuals to understand the world despite lacking optical perception.
2. The experimental results indicate that training with the proposed vision-centric text data is beneficial, leading to improved model performance on tasks like visual classification, open vocabulary detection, and VQA.

**Weaknesses:**

1. **Poor presentation**:

The paper mainly consists of two parts: the first is how to construct an instruction-tuning dataset, and the second is how the instruct-tuned LLMs can assist multimodal models.

**(1)** For the first part, the authors should pose some cases from the text instruction dataset in the main body of the paper rather than relegating them to the appendix. Otherwise, only through reading section 3, I can hardly understand what kind of data the authors aim to curate or why the curated data can achieve the authors’ goal.

**(2)** For the second part, the authors propose two ways to leverage the tuned LLMs. The second way is multimodal LLM, which is more intuitive and aligns with current prevalent methods. However, for the first way, i.e., LLM assisting vision models, it cost me a lot of time to figure out how the LLM helps visual classification and detection. A diagram illustrating this process would enhance clarity.

**(3)** Many of the expressions in the paper are irregular. For example, the notation ‘→’ used in Table 1 (M-7B → Mistral-7B) lacks clarity, and the actual name of the test dataset is not labeled in the caption of Table 7.

2. **Comparative Analysis**:

While I appreciate the motivation, I wonder which data for learning is more efficient and effective: vision-centric text data or vision-text data. Could the authors design an experiment to compare these two approaches?

For example, in Table 2, if I understand correctly, the authors utilize Mistral-7B fine-tuned with vision-centric text data. What would happen if Mistral-7B were fine-tuned with vision-text data of the same volume?

3. **Inefficiency of Fine-SFT**:

The method of fine-grained supervised fine-tuning (Fine-SFT) appears inefficient, as it necessitates calculating additional token weights.

**Questions:**

1. What is the computational cost associated with calculating the additional token weights in Fine-SFT?

---

### Official Review · Reviewer_9DdE · 2024-11-02

**Soundness:** 3
**Presentation:** 3
**Contribution:** 2
**Rating:** 5
**Confidence:** 5

**Summary:**

This paper proposes BRAILLEVISION-360K, which is a vision centric text instruction datasets constructed  from three aspects: perception,  abstraction and reasoning.   Experimental results show that text-based instruction fine-tuning with BRAILLEVISION-360K can improve the vision-centric skills for LLMs.

**Strengths:**

1. The paper is well written and easy to understand
2. The topic is interesting by exploring text knowledge to improve visual ability.
3. Experiments are good on some benchmarks.

**Weaknesses:**

1. What I am concerned about is the text performance. Will the method proposed in this paper hurt the text capability of LLM?

2. Does vicuna contain the same amount of data in BrailleVision? If not, the experiment is unfair.

3. Most multimodal benchmarks are in-domain or traditional VQA. Why not validate on the latest MLLM benchmarks like MMbench and MMVet, which can better reflect the effectiveness of the method.

4. typos：Line 52 and 84 are missing a space

**Questions:**

See weakness

---

### Official Review · Reviewer_dhFQ · 2024-11-03

**Soundness:** 2
**Presentation:** 2
**Contribution:** 2
**Rating:** 3
**Confidence:** 4

**Summary:**

The paper investigates an interesting approach: improving visual capabilities of Vision-Language Models (VLMs) through text-only training. A large-scale textual instruction tuning dataset featuring visual-related capabilities (e.g., classification, video summarization, and Visual Question Answering) is constructed. The authors empirically show that supervised fine-tuning (SFT) on this dataset can increase downstream performance on VLM benchmarks.

**Strengths:**

The idea of learning visual capabilities without visual data is compelling. Through solid and extensive experiments on a variety of datasets and benchmarks, the authors demonstrate the effectiveness of their method.

**Weaknesses:**

1. It is not clearly presented what exactly the "visual skills" learned through text-only training are. It appears more like learning and fitting the input-output format and boosting instruction-following abilities in visual benchmarks, rather than actual perceptual abilities. The core challenge in visual tasks—perception, i.e., extracting semantic information from raw pixels—seems untouched, while task format and instruction following capabilities can be well learned through NLP instruction dataset.

2. The additional text-only training requires extra computation and annotated datasets. I question whether allocating an equivalent amount of computation for visual instruction tuning would yield more substantial improvements. Incorporating the visual datasets used for BrailleVision-360k generation (e.g., ImageNet, Ego4D, VQAv2) directly as visual instruction tuning data might also lead to significant performance enhancements.

3. Generating the BrailleVision-360k dataset is complex and requires several additional steps and dependencies (e.g., CLIP, BLIP). A simpler baseline could be considered and compared to verify the necessity of the proposed method: translating images in visual instruction tuning datasets (e.g., LLaVa 1.5 dataset) into captions to derive a text-only dataset. This baseline is more straightforward and direct, and it would be simpler to implement.

4. Writing and Typo Suggestions
- **Line 048**: "supervised finetuning with instruction following data (IFT)" should be revised to "Supervised fine-tuning (SFT)," which are more commonly used terms. In modern LLMs, an alignment stage (e.g., RLHF or DPO) is often also included.
- **Line 084**: A space is missing between two sentences.
- **Line 097**: The term "semantic knowledge" is unclear.

**Questions:**

N/A

---

### Meta-Review · Area_Chair_FmMA · 2024-12-19

**Metareview:**

This work introduced a new instruction tuning dataset called BrailleVision, aimed at improving the vision-related capabilities for LLMs. The authors proposed an interesting way of contructing pure text instruction data but covers various vision tasks such as perception, summarization and spatial reasoning. It turned out that the after the instruction tuning, the LLMs can gain better vision performance. The authors further applied this vision-centric LLMs for image classification, object detection and multimodal LLM tasks, and showed superior performance to the baselines, respectively.

As many other reviewers, the ACs think the idea of leveraging pure text data to enhance the vision-centric capability for LLMs is appealing. This differs from most conventional way of curating image-text paired data to improve the performance in a straightforward way.

However, when reading through the submission and the reviews by all reviewers, the ACs agreed with the reviewers that this proposed method unfortunately is not supported by solid executiong and experiments, as well as comprehensive analysis on top of the curated dataset. To the end, all reviewers gave negative ratings, while the authors did not response to any.

To conclude, the ACs think the proposed method in this work is interesting and novel, but the concrete implementation for this work is relatvely much poor. We highly encourage the authors take into account the reviewers' comments to polish the submission.

**Additional Comments On Reviewer Discussion:**

It is unfortunate that the authors did not attempt to address the reviewers' concern during rebuttal session. As such all raised concerns by the reviewers are not addressed.

---

### Decision · Program_Chairs · 2025-01-22

Reject